

# We're building it up to burn it down: fire occurrence and fire-related climatic patterns in Brazilian biomes

Luisa Maria Diele Viegas[1,2], Lilian Sales[3], Juliana Hipólito[1,4], Claudjane Amorim[5], Eder Johnson de Pereira[6], Paulo Ferreira[7,8,9], Cody Folta[10], Lucas Ferrante[4], Philip Fearnside[4], Ana Claudia Mendes Malhado[5,11], Carlos Frederico Duarte Rocha[12] and Mariana M. Vale[13]

[1] Instituto de Biologia, Universidade Federal da Bahia, Salvador, Bahia, Brazil
[2] Fórum Clima Salvador, Salvador, Brazil
[3] Department of Biology, Faculty of Arts and Science, Concordia University, Montreal, Canada
[4] Instituto Nacional de Pesquisas da Amazônia, Manaus, Amazonas, Brazil
[5] Universidade Federal de Alagoas, Maceió, Alagoas, Brazil
[6] Instituto Federal do Maranhão, Bacabal, Maranhão, Brazil
[7] Research Center for Endogenous Resource Valorization, Portalegre, Portugal
[8] Department of Economic Sciences and Organizations, Portalegre Polytechnic Institute, Portalegre, Portugal
[9] Center for Advanced Studies in Management and Economics, Institute for Research and Advanced Training, Universidade de Evora, Evora, Portugal
[10] Department of Biology, University of Maryland at College Park, College Park, MD, United States of America
[11] Research Center in Biodiversity and Genetic Resources, University of Porto, Vairao, Portugal
[12] Departamento de Ecologia, Instituto de Biologia, Universidade do Estado do Rio de Janeiro, Rio de Janeiro, Brazil
[13] Universidade Federal do Rio de Janeiro, Rio de Janeiro, Brazil

Corresponding author
Luisa Maria Diele Viegas, luisa.mviegas@gmail.com

## ABSTRACT

**Background.** Terrestrial biomes in South America are likely to experience a persistent increase in environmental temperature, possibly combined with moisture reduction due to climate change. In addition, natural fire ignition sources, such as lightning, can become more frequent under climate change scenarios since favourable environmental conditions are likely to occur more often. In this sense, changes in the frequency and magnitude of natural fires can impose novel stressors on different ecosystems according to their adaptation to fires. By focusing on Brazilian biomes, we use an innovative combination of techniques to quantify fire persistence and occurrence patterns over time and evaluate climate risk by considering key fire-related climatic characteristics. Then, we tested four major hypotheses considering the overall characteristics of fire-dependent, fire-independent, and fire-sensitive biomes concerning (1) fire persistence over time; (2) the relationship between climate and fire occurrence; (3) future predictions of climate change and its potential impacts on fire occurrence; and (4) climate risk faced by biomes.

**Methods.** We performed a Detrended Fluctuation Analysis to test whether fires in Brazilian biomes are persistent over time. We considered four bioclimatic variables whose links to fire frequency and intensity are well-established to assess the relationship between climate and fire occurrence by confronting these climate predictors with a fire occurrence dataset through correlative models. To assess climate risk, we calculated the

climate hazard, sensitivity, resilience, and vulnerability of Brazilian biomes, and then we multiplied the Biomes' vulnerability index by the hazards.

**Results**. Our results indicate a persistent behaviour of fires in all Brazilian biomes at almost the same rates, which could represent human-induced patterns of fire persistence. We also corroborated our second hypothesis by showing that most fire-dependent biomes presented high thermal suitability to fire, while the fire-independent biome presented intermediate suitability and fire-sensitive biomes are the least suitable for fire occurrence. The third hypothesis was partially corroborated since fire-dependent and independent biomes are likely to increase their thermal suitability to fire, while fire-sensitive biomes are likely to present stable-to-decreasing thermal suitability in the future. Finally, our fourth hypothesis was partially corroborated since most fire-dependent biomes presented low climate risk, while the fire-independent biome presented a high risk and the fire-sensitive biomes presented opposite trends. In summary, while the patterns of fire persistence and fire occurrence over time are more likely to be related to human-induced fires, key drivers of burned areas are likely to be intensified across Brazilian biomes in the future, potentially increasing the magnitude of the fires and harming the biomes' integrity.

**Subjects** Conservation Biology, Ecology, Coupled Natural and Human Systems, Climate Change Biology, Environmental Impacts

**Keywords** Wildfires, Fire persistence, Climate hazard, Sensitivity index, Resilience, Vulnerability, Climate risk

# INTRODUCTION

The world is predicted to reach an irreversible climate tipping point if average global temperature exceeds 1.5 °C above pre-industrial levels (*Ripple et al., 2019*; *IPCC, 2021*). Terrestrial biomes in South America are likely to experience a persistent increase in environmental temperature, often combined with moisture reduction and changes in wind patterns (*Anjos et al., 2021*; *Burton et al., 2022*). In addition, natural wildfire ignition sources, such as lightning, can become more frequent under climate change scenarios, since favorable conditions are likely to occur more often (*Clark, Ward & Mahowald, 2017*; *Krasovska, Buravchenko & Tsviashchenko, 2018*). Such conditions include fuel availability, a flammable mixture of organic compounds, and cloud cover, which can change to increase the frequency of storm clouds bearing electric charge (*Krasovska, Buravchenko & Tsviashchenko, 2018*). As key drivers of naturally burned areas (*Burton et al., 2022*), changes in these characteristics are prone to affect fire occurrence, with fire intensity and spread likely to increase after ignition depending on local weather conditions (*Clark, Ward & Mahowald, 2017*; *Podschwit et al., 2018*; *Li et al., 2022*).

Persistent fires in fire-dependent biomes are likely to occur regularly under natural conditions, and local biodiversity should be adapted to those conditions (*Pausas & Bradstock, 2007*). However, along with natural fire ignition sources, farmers commonly use human-induced fires as a management tool to clear new areas for settlements, ranching, agriculture and logging (*Brunel et al., 2021*). Such practices have been used in Brazilian

grass-dominated areas for centuries (*Pivello, 2011*), usually aiming to remove excessive dead biomass during the dry season and stimulate the regrowth of grasses with high nutritional value for grazing animals (*Van der Werf et al., 2008*; *Da Silva Junior et al., 2020*; *Brunel et al., 2021*). Although this is an effective practice for improving the productivity or ranching (*Laterra et al., 2003*), its inadequate application can decrease system resilience (*Roberts, 2000*).

Human-induced fires are likely to be more intense than natural fires and can have different effects according to the system's fire susceptibility (*Da Silva Junior et al., 2020*). In a fire-independent or fire-sensitive biome, where fires are not likely to occur naturally and therefore are not persistent, emergency fire plans must be designed considering the local potential for fire spread, which can be assessed by analyzing local environmental characteristics (*Santos et al., 2021*). Similarly, landscape alterations must also be acknowledged in fire-dependent biomes, but strategies for fire control should consider the ecological and cultural roles of fire in the landscape (*Santos et al., 2021*).

Whether or not fire persists over time, it occurs in all Brazilian biomes (*Da Silva Junior et al., 2020*). Understanding the main sources of fire ignition (*i.e.*, natural or human-induced) and the persistence patterns of fire is therefore vital to developing adequate fire-management policies and to avoiding the loss of biodiversity and ecosystem services (*Roberts, 2000*). It is important to characterize each biome's fire-related climate risk resulting from the interaction between hazard, exposure, and vulnerability (*sensu IPCC, 2018*; see *Foden et al., 2019*). "Hazard" refers to the potential occurrence of climate-related events or trends that may harm the system, while "exposure" is the presence of the system in places that are potentially affected (*Foden et al., 2019*). The "Vulnerability" is its propensity or predisposition of a system to be adversely affected and has many components, including sensitivity and resilience (*Foden et al., 2019*). "Sensitivity" can be described as the degree to which a system is affected by climate change, while "resilience" is the capacity of the system to cope with disturbance and keep its essential functions and structure (*IPCC, 2021*). Resilience reflects the system's ability to maintain its adaptation and transformation capacities, which can be assessed by quantifying vegetation loss and protected areas (PAs) (*IPCC, 2021*).

Changes in climate risk, along with fire occurrence and persistence patterns over time, can trigger significant modifications in ecosystem structure and internal feedbacks and can disrupt ecological functions, affecting biodiversity and human livelihoods (*Anjos et al., 2021*; *Diele-Viegas, 2021*). However, knowledge gaps on the specificities of these characteristics in Brazilian biomes prevent the development of adequate management policies to minimize fire impacts.

Here we examine fire persistence and occurrence patterns over time and evaluate climate risk by considering key fire-related climatic characteristics in Brazilian biomes, anticipating fire occurrence under different climate change scenarios through an innovative combination of techniques. We test four hypotheses concerning the overall characteristics of fire-dependent, fire-independent, and fire-sensitive biomes: the first is related to fire persistence over time; the second approaches the relationship between climate and fire occurrence; the third focuses on future predictions of climate change and its potential
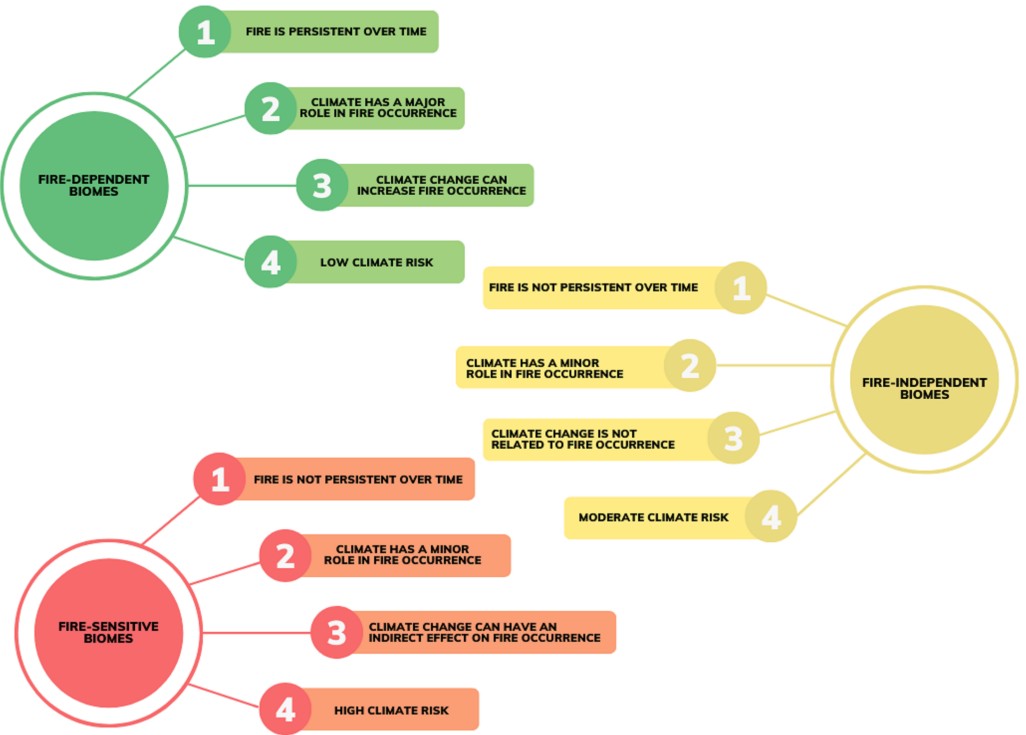

**Figure 1** **Hypothetic patterns of fire persistence, fire occurrence, and climate risk in Brazilian fire-dependent (Cerrado, Pampa, and Pantanal), fire-independent (Caatinga) and fire-sensitive (Amazon and Atlantic Forest) biomes.** Climate risk is predicted to be low in fire-dependent biomes due to their potential low hazard associated with low sensitivity and intermediary resilience. For fire-independent biomes, climate risk is expected to be moderate since it is likely to present a high hazard related to high sensitivity and low resilience. Finally, fire-sensitive biomes are likely to present high climate risk due to an intermediate-to-high hazard and high sensitivity, even considering their potentially high resilience.

impacts on fire occurrence, and the last focuses on the climate risk faced by the biomes (Fig. 1).

## MATERIALS & METHODS

Brazil is among the most biodiverse countries globally and plays a major role in regulating the South American climate system, mainly through the evapotranspiration of the Amazon forest (*Convention on Biological Diversity, 2021*). It also hosts important hotspots of biological diversity that support diverse ecosystem services (*Jenkins & Schaap, 2018*). The six Brazilian biomes (Amazon, Atlantic Forest, Caatinga, Cerrado, Pampa, and Pantanal) hold large carbon stocks in their forests and soils, besides having the largest freshwater reserve in the world (*Souza et al., 2020*).

Brazilian grassland, savanna and wetland biomes (*i.e.*, Cerrado, Pampa, and Pantanal) are fire-dependent; they have coevolved with lightning-driven fires and benefit from seasonal fires (*Hardesty, Myers & Fulks, 2005*; *Pivello et al., 2021*). On the other hand, the semi-arid scrub forests of the Caatinga are fire-independent, since climatic conditions are not favorable to fire occurrence (*e.g.*, there is a low incidence of lightning events), and

the system lacks enough biomass to carry fire (*Hardesty, Myers & Fulks, 2005*; *Pivello et al., 2021*). Finally, humid tropical forests (*i.e.*, Amazon and Atlantic Forest) are not adapted to fire and are therefore fire sensitive (*Hardesty, Myers & Fulks, 2005*; *Pivello et al., 2021*). We tested our four hypotheses considering the overall characteristics of fire-dependent, fire-independent, and fire-sensitive biomes, through the methods described below.

## Persistent fire behavior

Detrended Fluctuation Analysis (DFA) is a powerful tool for evaluating long-range dependence in individual time series, which could be applied to identify and measure the existence of autocorrelation in the context of non-stationary time series (*Peng et al., 1994*). It can assess the extent to which trends in fire events observed in the past imply the maintenance of that behavior in the future, thus evaluating whether fire events are random or persistent over time (*Peng et al., 1994*; *Tong et al., 2019*; *Murari et al., 2020*).

We performed a DFA by considering a time-series of fire occurrences from the reference satellite (AQUA_M-T: MODIS sensor, early afternoon pass), downloaded from the website of the Brazilian National Institute for Space Research (*INPE, 2020*, p. 202). The data have 1-km × 1-km pixel spatial resolution, depicting fires from November 2011 to October 2020. The DFA was calculated for a 10-year time series with $t$ equidistant observations. The first step of the analysis consisted of calculating the fire profile:

$$X_t = \sum_{i=1}^{t} (x_i - \langle x \rangle).$$

where the original time series $x_i$ is the fire occurrence per year, with $i = 1, \ldots, N$, and N is the total number of measurements recorded, and $\langle x \rangle$ is the average observed fire occurrence. This profile was then divided into mutually exclusive boxes of equal dimensions $s$ (the considered timescale; $N/s$), and a local trend was calculated using ordinary least squares to detrend the profile:

$$X_s(t) = X_t - z(t).$$

where $z(t)$ is the polynomial fit for the respective follow-up. Finally, the DFA function was calculated for all $s$ values according to the following equation:

$$F(s) = \sqrt{\frac{1}{N} \sum_{t=1}^{N} \left( X_{(s)}(t) \right)^2}.$$

The log–log regression was obtained between $F(s)$ and $s$, resulting in a power-law given by

$$F(n) \propto n^{\alpha}.$$

Through the $\alpha$ exponent obtained from the DFA, it is possible to assess the extent to which the trend observed in the past time series implies the maintenance of that behavior in the future, indicating a long-term memory effect in the series. Non-correlated series are expected to return $\alpha = 0.50$ and represent a typical case of a random walk. These series are likely to show long-range persistence when $\alpha > 0.50$ and anti-persistent behavior when

$\alpha$ <0.50. Although opposite precipitation anomalies are expected for NE and SE Atlantic Forest (*Reboita et al., 2022*), we did not separate the Atlantic Forest into NE and SE regions for this analysis because DFA does not consider precipitation patterns in its equation.

## Climate and fire occurrence

Confronting data of climatic predictors with fire occurrence can be useful for understanding the particularities of this relationship in different biomes and their consequences under different climate-change scenarios. Correlative models associating environmental (*e.g.*, climatic predictors) and geographic (*e.g.*, georeferenced occurrence points) spaces have been widely used in conservation biology to predict the potential occurrence area of a species under different environmental conditions (*Guisan & Zimmermann, 2000*).

By assuming that climatic conditions are at least partially responsible for fire occurrence, we adapted this method to model fire occurrence by highlighting climatically similar regions where fires were recorded, thus predicting the probability of fire occurrence through time and over geographic space. A synthesis of the modeling steps can be found in Table S1.

To assess the relationship between climate and fire occurrence, we considered four bioclimatic variables whose links to fire frequency and intensity are well established (*Oliveira-Júnior et al., 2020*): two annual variables (annual mean temperature; BIO1, and annual precipitation; BIO12) and two variables related to the dry (or fire-prone) season (mean temperature of the driest quarter; BIO9, and precipitation of the driest quarter; BIO17). Climatic raster files with historical data (average of years 1970-2000) were obtained from WorldClim version 2.1 (*Fick & Hijmans, 2017*) at 2.5 arc-minute resolution.

Climatic predictors were confronted with the fire-occurrence dataset available from INPE (*INPE, 2020*). Each fire source registered from 2002 to 2020 was considered a ''fire occurrence'', and its georeferenced location was extracted, totaling 4,546,557 occurrence points. To reduce the effects of sampling bias, we filtered the occurrence dataset to ensure that localities were at least 10 km distant of one another, besides removing duplicate coordinates and any point falling outside Brazil's geographic limits, resulting in 4,463,071 valid points. In addition, we randomly subsampled 1% of the occurrence points ($N = 4463$) to proceed with the analysis.

The climatic conditions associated with each location were assessed with correlative models that related environmental characteristics to fire occurrences. We adapted this method to fire occurrence, assuming that climate directly influences these phenomena. We therefore caution that this method assumes that climate conditions are at least partially responsible for fire occurrence.

The modelled relationship between climate and fire was projected into geographic space to highlight climatically similar regions and relate them to the recorded fires in Brazil. By assessing the environmental conditions associated with fire events, it was possible to predict the probability of fire occurrence through time and over geographic space. We fitted models using three algorithms: bioclimatic envelopes (BIOCLIM), generalized linear models (GLM), and support vector machines (SVM). We randomly sampled 10,000 pseudoabsences for the BIOCLIM and GLM algorithms while maintaining occurrence prevalence for SVM (*Barbet-Massin et al., 2012*). A final consensus map was obtained by

weighting the cell-based prediction of fire probability by the accuracy of the parent model, as explained below.

We evaluated the accuracy of our model output with a sub-sampling procedure where 30% of fire-occurrence records measured the performance of models fitted to the remaining 70% of the records. We ran ten replicates for each algorithm (in a total of 30 models) to increase the robustness of the results. The best models were selected based on an Area Under the Curve (AUC) over 0.7 (*Hanley & McNeil, 1982*), True Skill Statistic (TSS) over 0.3, and Threshold over 0.8. The threshold was calculated as the point at which the sum of the sensitivity (true positive rate) and specificity (true negative rate) was highest (*Hijmans et al., 2017*; *Shabani, Kumar & Ahmadi, 2018*). Based on the combination of these three accuracy methods, we selected five SVM replicas to proceed with the analyses (see Table S2).

We projected our consensus models of fire occurrence based on climate forecasts to predict future fire risk. We projected fire occurrence for two time periods (2041-2060, hereafter 2050; and 2081-2100, hereafter 2090) according to two shared socioeconomic pathways (SSP) (*Riahi et al., 2017*), SSP2 45 and SSP5 85 (*IPCC, 2021*). The future climate projections considered the ensemble (average) of the three general circulation models (GCMs) of the Coupled Model Intercomparison Project Phase Six (CMIP6) (*Eyring et al., 2016*) with particularly good performance in South America (*Cannon, 2020*): (1) Beijing Climate Center model (BCC-CSM2-MR); (2) Institute Pierre Simon Laplace model (IPSL-CM6A-LR); and (3) Model for Interdisciplinary Research on Climate (MIROC6).

The geographic boundaries of the Brazilian biomes were downloaded from the Brazilian Institute of Geography and Statistics (IBGE, http://ibge.gov.br/). We separated the Atlantic Forest into its northeastern (NE), and southeastern (SE) portions since the predicted changes in precipitation have opposite signs in these areas: reduction in precipitation in the NE and increase in the SE (*Brazilian Panel on Climate Change (PBMC) fix order in list, 2014*; *Reboita et al., 2022*). The analysis was performed using the *biomod2* (*Thuiller et al., 2020*), *dismo* (*Hijmans et al., 2017*), *raster* (*Hijmans, 2016*), and *rgdal* (*Bivand, Keitt & Rowlingson, 2020*) packages in R 3.5.1 software (*R Core Team, 2020*).

## Climate risk

We assessed Brazilian biomes' climate risk through metrics of climate hazard, sensitivity and resilience (*Foden et al., 2019*; *IPCC, 2021*, p. 20). For climate hazard, we first calculated the percentage of change in bioclimatic values between future predictions and the present (see Table S3). We considered the same four bioclimatic variables used to assess the relationship between climate and fire occurrence (BIO1, BIO9, BIO12, and BIO17) at the same resolution (2.5 arc-minutes spatial resolution). Future predictions were based on SSP2 45 and SSP5 85 for two time periods (2050 and 2090) and considered the ensemble of the three GCMs mentioned above (BCC-CSM2-MR, IPSL-CM6A-LR, and MIROC6). Accordingly, the geographic boundaries of the Brazilian biomes were the same as those used to assess the relationship between climate and fire occurrence (IBGE, http://ibge.gov.br/). We then adapted the Regional Climate Change Index (RCCI) developed by *Giorgi (2006)* to calculate the climate hazard of the fire season in Brazilian biomes. This comparative index was developed to identify "hotspots" of climate change (*Giorgi, 2006*). The RCCI is

**Table 1** *n* **values considered in the definition of RCCI, based on** *Giorgi (2006).* ΔP is the percentage of change in annual precipitation; ΔσP is the interannual variability of precipitation; ΔσT is the interannual variability of temperature; and RWAF is the regional warming amplification factor.

| *n* | ΔP (%) | ΔσP (%) | RWAF | ΔσT (%) |
|---|---|---|---|---|
| 0 | <5 | <5 | <1.1 | <5 |
| 1 | 5–10 | 5–10 | 1.1–1.3 | 5–10 |
| 2 | 10–15 | 10–20 | 1.3–1.5 | 10–15 |
| 4 | >15 | >20 | >1.5 | >15 |

defined as:

$$\text{RCCI} = [n(\Delta P) + n(\Delta\sigma_P) + n(\text{RWAF}) + n(\Delta\sigma_T)]_{\text{WS}} + [n(\Delta P) + n(\Delta\sigma_P) + n(\text{RWAF}) + n(\Delta\sigma_T)]_{\text{DS}}.$$

where *n* is an empirical factor that depends on the magnitude of the change (Table 1); $\Delta P$ is the percentage of change in BIO12 (annual precipitation) recovered for each biome; $\Delta\sigma_P$ is the interannual variability of precipitation; $\Delta\sigma_T$ is the interannual variability of temperature; and RWAF is the regional warming amplification factor, *i.e.*, the difference between the change in BIO1 (annual mean temperature) recovered for each biome and the mean global temperature change (2 °C and 2.7 °C in SSP2 45 and 2.4 °C and 4.4 °C in SSP5 85, considering 2050 and 2090, respectively (*IPCC, 2021*)). Note that the original RCCI performs these calculations for both the wet (WS) and dry (DS) seasons. However, since we focused our analysis on the dry season (the fire-prone season), we considered $\text{RCCI}_{\text{DS}} = n(\Delta P) + n(\Delta\sigma_P) + n(\text{RWAF}) + n(\Delta\sigma_T)$ as a proxy for climate hazard, which was calculated for each biome in each scenario and time period (see Table S4).

The sensitivities of Brazilian biomes were estimated based on the Vegetation Sensitivity Index (VSI) (*Seddon et al., 2016*; see Table S5). This index is a useful method to quantify ecosystem sensitivity based on the relative variance of vegetation productivity compared to three ecologically important MODIS-derived climate variables: air temperature, water availability and cloud cover (*Seddon et al., 2016*). Since these three climatic variables are key drivers of burning, this index can also indicate whether the environment would be sensitive to fire-related climatic conditions. The comparisons are made for each 5-km grid square for the months in which EVI and climate are found to be related. We calculated the sensitivity of the Brazilian biomes by using the *raster* (*Hijmans, 2016*, p. 201) and *rgdal* (*Bivand, Keitt & Rowlingson, 2020*) packages in R 3.5.1 software (*R Core Team, 2020*).

We calculated the Biomes' resilience based on vegetation loss and the area outside PAs (see Table S5). Natural vegetation is an essential regulator of ecosystem services, and its destruction has been considered to be the leading cause of species extinction (*Gonçalves-Souza, Verburg & Dobrovolski, 2020*; *Gonçalves-Souza et al., 2021*). On the other hand, PAs are the cornerstone strategy for biodiversity conservation, playing a major role in the maintenance of ecosystem services and adequate environmental conditions for the survival of local species (*Bernard, Penna & Araújo, 2014*). Although Brazil has the largest PA network in the world, covering over 250 million ha and 29.4% of the country's territory (*UNEP-WCMC & IUCN, 2022*), its coverage is not proportionally distributed

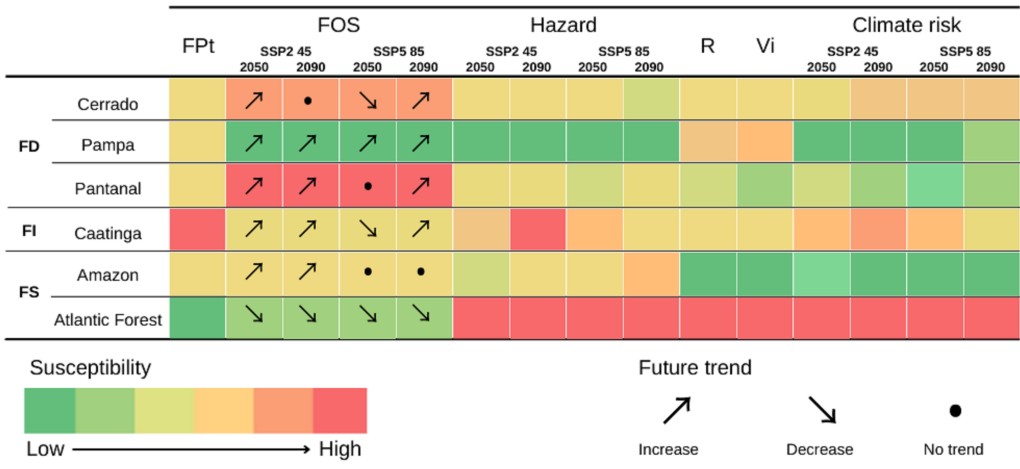

**Figure 2** **Fire Persistence over time (FPt), Fire Occurrence Suitability (FOS), Climate Hazard, Resilience (R), Vulnerability Index (Vi) and Climate risk of fire-dependent (FD), fire-independent (FI), and fire-sensitive (FS) Brazilian biomes.** Colors represent aspects that make them more (red) or less (green) susceptible to that feature than the other biomes. Arrows indicate the future trends of that feature in different climate change scenarios.

among the Brazilian biomes (*Oliveira et al., 2017*). Vegetation loss was calculated from current land-use and land-cover data (*IBGE, 2020*; see *Fernandes et al., 2017*), while the area outside PAs was measured based on georeferenced data from the Chico Mendes Institute for Biodiversity (*ICMBio, 2020*). We considered the resilience status to be the arithmetic mean of these two indicators, where the lower the mean value, the greater the resilience status (*Lapola et al., 2020*).

We then used the sensitivity and resilience metrics to determine the biomes' vulnerability to climate change. For this, we compared the sensitivity and resilience of each biome and calculated the relative weights for the lower sensitivity and resilience as the difference between the sensitivity/resilience of the biome and the lower sensitivity/resilience value found between biomes. We considered the vulnerability index of each biome to be the arithmetic mean of the relative weight of sensitivity ($\Delta$S) and the relative weight of resilience ($\Delta$R). We assessed climate risk by multiplying the biomes' vulnerability index by the hazards per SSP and year (adapted from *Foden et al., 2019*). We summarized our findings concerning persistent fire behavior, fire occurrence, climate hazard, resilience, vulnerability, and climate risk of Brazilian biomes in Fig. 2 (see Table S6 for details).

## RESULTS

### Persistent fire behavior

Over the last ten years, fires in Brazil occurred mainly in the Amazon and Cerrado biomes (Fig. 3). While 2013 and 2018 had the lowest fire occurrences in the decade, 2012 and 2020 had the highest occurrences (Fig. 3). The Pantanal and Pampa biomes were particularly affected in 2020 when they suffered from the highest fire occurrences in the decade (Fig. 3).

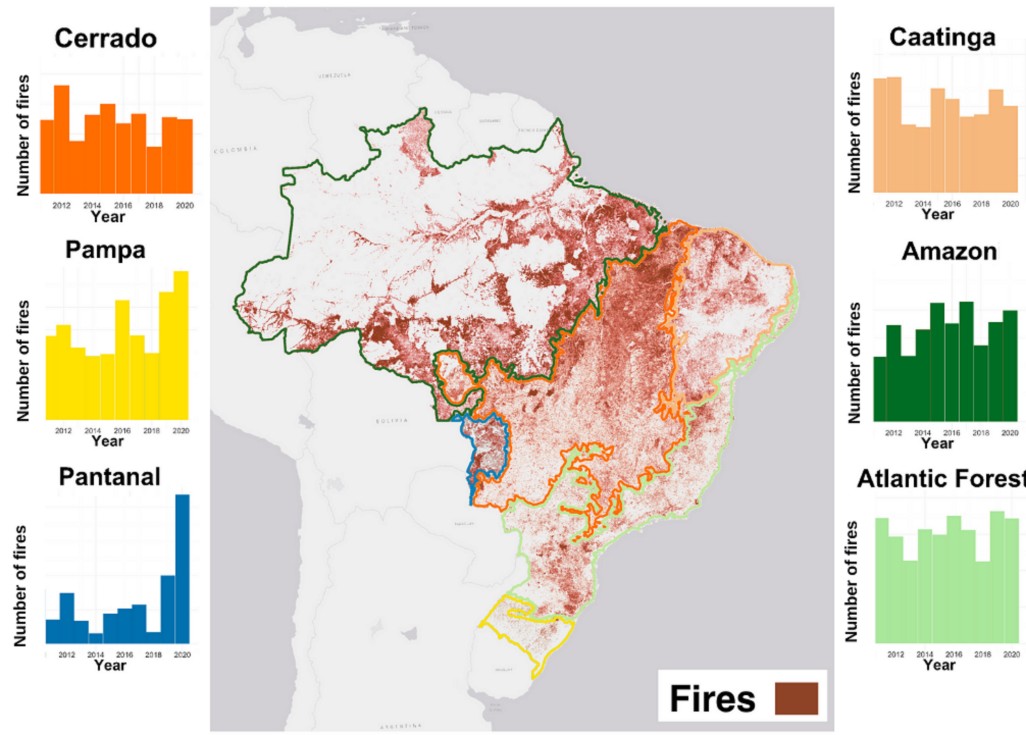

**Figure 3 Fire persistence in Brazilian biomes between 2010 and 2020.** Source: *INPE (2020)*. Brown points in the map are the georeferenced location of fire sources during the analyzed period. Bar plots indicate the distribution of these fire sources over the years per biome.

**Table 2 Detrended fluctuation analysis (DFA) per Brazilian biome.**

| NFDC | Biome | DFA ($\alpha \pm$ SD) |
|---|---|---|
| | Cerrado | $0.63 \pm 0.07$ |
| Fire-Dependent | Pampa | $0.63 \pm 0.06$ |
| | Pantanal | $0.63 \pm 0.05$ |
| Fire-Independent | Caatinga | $0.61 \pm 0.06$ |
| Fire-Sensitive | Amazon | $0.63 \pm 0.07$ |
| | Atlantic Forest | $0.72 \pm 0.07$ |

Notes.

NFC, Natural-Fire Dependence Classification; SD, Standard Deviation.

Our results indicate a persistent behavior of fires in all Brazilian biomes, with small differences among natural-fire dependence classifications (Table 2, Fig. 3). Fire-dependent biomes had the same persistence behavior as the Amazon rainforest ($\alpha = 0.63$), but only Cerrado presented the same variation (SD $= 0.07$). The least persistent behavior was in the Caatinga ($\alpha = 0.61 \pm 0.06$), the only fire-independent biome. The Atlantic Forest showed the most persistent fire behavior over the years and therefore is projected to have large amounts of future fire despite being fire sensitive (Table 2, Fig. 3).

**Table 3  Mean ± Standard Deviation of thermal suitability of the Brazilian biomes (%) to fire occurrence considering different scenarios for climate change (SSP2 45 and SSP5 85) and different years (2050 and 2090).**

| NFDC | Biome | Present | 2050 SSP2 45 | 2090 SSP2 45 | 2050 SSP5 85 | 2090 SSP5 85 |
|---|---|---|---|---|---|---|
| Fire-Dependent | Cerrado | 0.78 ± 0.23 | 0.80 ± 0.23 | 0.78 ± 0.23 | 0.77 ± 0.23 | 0.80 ± 0.22 |
| | Pampa | 0.19 ± 0.09 | 0.23 ± 0.08 | 0.25 ± 0.10 | 0.23 ± 0.09 | 0.23 ± 0.09 |
| | Pantanal | 0.88 ± 0.10 | 0.90 ± 0.08 | 0.89 ± 0.08 | 0.88 ± 0.08 | 0.89 ± 0.08 |
| Fire-Independent | Caatinga | 0.64 ± 0.28 | 0.66 ± 0.28 | 0.66 ± 0.28 | 0.63 ± 0.27 | 0.65 ± 0.28 |
| | Amazon | 0.62 ± 0.34 | 0.63 ± 0.34 | 0.63 ± 0.33 | 0.62 ± 0.33 | 0.62 ± 0.34 |
| Fire-Sensitive | Atlantic Forest | 0.49 ± 0.27 | 0.50 ± 0.24 | 0.47 ± 0.23 | 0.46 ± 0.23 | 0.45 ± 0.22 |
| | Atlantic Forest NE | 0.48 ± 0.25 | 0.51 ± 0.21 | 0.46 ± 0.21 | 0.46 ± 0.21 | 0.43 ± 0.19 |
| | Atlantic Forest SE | 0.50 ± 0.28 | 0.49 ± 0.26 | 0.48 ± 0.25 | 0.46 ± 0.25 | 0.47 ± 0.25 |

**Notes.**

NFDC,  Natural-Fire Dependence Classification.

## Climate and fire occurrence

The current thermal suitability of the Brazilian biomes to fire occurrence varied from 19% in the Pampa to 88% in the Pantanal (Table 3). Most fire-dependent biomes presented high thermal suitability to fire, while the fire-independent biome presented an intermediate suitability and fire-sensitive biomes are the least suitable to fire occurrence (Table 3).

Although the Pampa is currently the least likely biome to burn, it presents the greatest proportional increase in thermal suitability considering future scenarios (Table 3). Thermal suitability to fire is also likely to be greater in the Amazon even considering an optimistic climate-change scenario (SSP2 45), but it tends to decrease by the year 2050 considering a more pessimistic scenario (SSP5 85; Table 3). The Northeastern Atlantic Forest tends to increase its thermal suitability by 2050 considering SSP2 45, while the Southeastern Atlantic Forest tends to decrease its thermal suitability in all forecasts. Caatinga, Cerrado, and Pantanal are likely to present increased thermal suitability to fires in most scenarios, except for SSP5 85 in the year 2050 (Table 3). Overall, fire-dependent and fire-independent biomes are likely to increase their thermal suitability, while fire-sensitive biomes are likely to present stable-to-decreasing thermal suitability to fire in the future (Table 3).

## Climate risk

While the Pampa had the lowest climate hazard in all evaluated scenarios, the Atlantic Forest presented the highest (Table 4). The other biomes varied in their compared positions, but, overall, fire-dependent biomes were usually among those with the lowest climate hazards, while the fire-independent biome presented the second highest hazard in almost all scenarios (Table 4).

Despite its low climate hazard, the Pampa is predicted to have the highest sensitivity among the Brazilian biomes (Table S5). On the other hand, the Pantanal had the lowest sensitivity to fire-related climatic conditions (Table S5). Fire-dependent biomes did not present a common pattern of sensitivity, although, on average, they are likely to present lower sensitivity than the fire-independent biome (17.3 ± 3.3; compared to 19.6 ± 3.7 for the Caatinga). Finally, contrary to expectations, fire-sensitive biomes figured among those with the lowest sensitivities to fire-related climatic conditions (Table S5).

**Table 4** Climate-related hazard, resilience and vulnerability of Brazilian biomes considering different scenarios for climate change (SSP2 45 and SSP5 85) and different years (2050 and 2090).

| NFDC | Biome | Hazard | | | | Resilience status | Vulnerability index |
|---|---|---|---|---|---|---|---|
| | | SSP2 45 | | SSP5 85 | | | |
| | | 2050 | 2090 | 2050 | 2090 | | |
| Fire-Dependent | Cerrado | 13.66 | 32.23 | 27.37 | 105.35 | 0.73 | 11.73 |
| | Pampa | 0.00 | 6.22 | 5.07 | 40.12 | 0.75 | 13.43 |
| | Pantanal | 14.54 | 32.98 | 22.28 | 112.04 | 0.67 | 6.40 |
| Fire-Independent | Caatinga | 16.96 | 49.40 | 36.26 | 124.61 | 0.70 | 10.87 |
| | Amazon | 8.20 | 33.89 | 29.88 | 139.95 | 0.54 | 1.33 |
| Fire-Sensitive | Atlantic Forest | 28.76 | 51.02 | 48.44 | 188.82 | 0.92 | 19.45 |
| | Atlantic Forest NE | 42.39 | 73.64 | 72.93 | 267.80 | 0.92 | 21.61 |
| | Atlantic Forest SE | 15.12 | 28.40 | 23.94 | 109.84 | 0.93 | 18.80 |

**Notes.**
NFDC, Natural-Fire Dependence Classification.

The most resilient biome was the Amazon rainforest (Table 4), which presented the lowest rates of vegetation loss added to the greatest proportional area under protection (Table S5). On the other hand, the Atlantic Forest and Pampa presented the highest rates of vegetation loss, besides being the least-protected biomes (Table S5), which led them to be considered the least resilient among the Brazilian biomes (Table 4). Consequently, the vulnerability of Brazilian biomes followed the same pattern as resilience, with the Amazon being the least vulnerable and the Atlantic Forest and Pampa being the most vulnerable biomes (Table 4).

Fire-dependent biomes vary from low to intermediate resilience and vulnerability, while fire-sensitive biomes present the opposite trends in both features (Fig. 2). Therefore, our results indicate no overall pattern of resilience or vulnerability among fire-dependent and fire-sensitive biomes. The fire-independent biome, in turn, presented the greatest resilience (0.70), but the highest vulnerability (10.87) compared to the averaged resilience and vulnerability of fire-dependent ($R = 0.72 \pm 0.4$; $Vi = 10.51 \pm 3.7$) and fire-sensitive ($R = 0.73 \pm 0.27$; $Vi = 10.39 \pm 12.8$) biomes.

The Atlantic Forest is the most at-risk biome in all evaluated scenarios, while the Amazon and Pampa are among the least at-risk (Table 5). Fire-dependent biomes presented an overall pattern of mid-to-low risk, while the fire-independent biome presented a higher climate risk (Table 5). No pattern was found for fire-sensitive biomes. On average, fire-dependent biomes presented the lowest climate risk in all evaluated scenarios, while the fire-sensitive biomes presented the highest risk for all but the predictions for 2090 considering the SSP2 45 (Table 5).

**Table 5  Climate risk of brazilian biomes considering their natural-fire dependence classification (NFDC).**

| NFDC | Biome | Climate risk | | | |
|---|---|---|---|---|---|
| | | SSP2 45 | | SSP5 85 | |
| | | 2050 | 2090 | 2050 | 2090 |
| Fire-Dependent | Cerrado | 160.16 | 377.90 | 320.91 | 1235.23 |
| | Pampa | 0.00 | 83.50 | 68.06 | 538.61 |
| | Pantanal | 93.06 | 211.07 | 142.59 | 717.07 |
| | Average ± SD | 84.41 ± 80.4 | 224.16 ± 147.6 | 177.19 ± 129.9 | 830.3 ± 361.8 |
| Fire-Independent | Caatinga | 184.36 | 536.98 | 394.15 | 1354.51 |
| | Amazon | 10.91 | 45.07 | 39.74 | 186.13 |
| Fire-Sensitive | Atlantic Forest | 559.14 | 992.08 | 941.82 | 3671.60 |
| | Average ± SD | 285.02 ± 387.6 | 518..58 ± 669.64 | 490.82 ± 637.93 | 1928.87 ± 2464.6 |
| | Atlantic Forest NE | 916.05 | 1591.36 | 1576.02 | 5787.16 |
| | Atlantic Forest SE | 284.26 | 533.92 | 450.07 | 2064.99 |

**Notes.**
SD, Standard Deviation.

## DISCUSSION

### Persistent fire behavior

Our first hypothesis focused on describing fire persistence in Brazilian biomes over time, and our results indicate a persistent behavior in all Brazilian biomes at almost the same rates. Such patterns would only be possible considering the human-induced fires to be occurring at a recurrent rate. In fact, the use of fire in Brazilian crop and pasture management has been well established for centuries (*Maezumi et al., 2018*), which creates a feedback loop, since repeatedly burned areas are more prone to new fires (*Hoffmann et al., 2020*). Our results therefore suggest human-induced patterns of fire persistence instead of natural patterns, which is an indication that human activities have already radically changed the natural fire regimes in Brazilian biomes with respect to the frequency and timing of burning (*Hardesty, Myers & Fulks, 2005*; *Pivello, 2011*).

Although we focused on fire frequency, other characteristics of fire regimes are also important when evaluating fire impacts on biodiversity and ecosystems. Specifically, the organic matter consumption by fire (*i.e.*, fire severity) can be used as a proxy for the energy output from fire (*i.e.*, fire intensity) and its impact on ecosystems (*Keeley, 2009*). In addition, changes in fire-season length (the period when fires are prone to spread due to climatic factors) can increase the occurrence of ignitions and the duration of burns, resulting in larger fires that are more likely to spread and negatively impact biodiversity (*Riley & Loehman, 2016*). Similarly, smoke emissions due to fires lead to reduced air quality, directly impacting human health and causing premature adult deaths that could be avoided (*Reddington et al., 2015*). Future studies on fire regimes in Brazilian biomes should focus on these multiple characteristics to improve our knowledge of Brazilian fire regimes and to increase the effectiveness of fire-management policies.

## Climate and fire occurrence

Our second hypothesis focused on understanding the climatic determinants of fire occurrence by assuming that climatic conditions are at least partially responsible for fire. Our results corroborated our expectations by showing that the most fire-dependent biomes presented high thermal suitability for fire, while the fire-independent biome presented an intermediate suitability and fire-sensitive biomes were the least suitable for fire occurrence. Similarly, *Oliveira et al. (2022)* also demonstrated that climate explains most of the fire variation in the Cerrado and Pantanal biomes, while land-use change explained most of the fire variation in the Amazon.

In our third hypothesis we considered possible tendencies of fire occurrence in the future considering different climate-change scenarios. Our results partially corroborated this hypothesis by showing that fire-dependent and independent biomes are likely to increase their thermal suitability for fire, while fire-sensitive biomes are likely to present stable-to-decreasing thermal suitability in the future. Specifically, thermal suitability for fire is likely to decrease in all scenarios for the Atlantic Forest but increase in the Amazon, considering the optimistic scenario for climate change (SSP2 45), while no trend is expected under the most severe scenario (SSP5 85). Extreme drought events are likely to become more frequent under different climate-change scenarios (*IPCC, 2021*), facilitating fire occurrence and spread. Prolonged drought events have already been recorded within the last decade. Lack of rainfall in 2019 and 2020 in the Pantanal, caused by the reduced transportation of humid air from the Amazon, led to a prolonged drought in this fire-dependent biome, which facilitated fire spread, culminating in the extreme fire event that burned over 30% of the biome in 2020 (*Mega, 2020*; *Libonati et al., 2020*; *Marengo et al., 2021*). Similarly, an increased drying effect in fire-sensitive biomes could result in increases in fire incidence, as predicted for the Amazon (*Aragão et al., 2018*).

## Climate risk

In our fourth hypothesis, we evaluated the climate risk of Brazilian biomes considering fire-related climatic variables and showed that biophysical conditions conducive to fires (increased temperature and decreased precipitation) are likely to affect all Brazilian biomes (see Table S3). However, as demonstrated in our fire occurrence models, these predicted changes could not produce the expected results on fire occurrence, which may seem contradictory at first: if flammable conditions are expected to become more widespread and fire frequency is somewhat predictable, fire occurrence should be expected to increase. However, such an assumption would only hold if the primary fire driver across biomes were the climate –which is not true, since human-induced fires are prevalent in all Brazilian biomes (*Pivello et al., 2021*).

Our results indicate no overall sensitivity pattern for fire-dependent biomes, and only the Pantanal followed our expectations of low sensitivity, while both Cerrado and Pampa were among the biomes that were most sensitive to the evaluated fire-related climatic conditions, together with the fire-independent biome (Caatinga). Contrary to expectations, fire-sensitive biomes presented the lowest vegetation sensitivities compared to other biomes. These tropical forests could be operating at different timescales with

regard to their expected sensitivity to potential precipitation thresholds identified in these systems (*Lenton et al., 2008*; *Seddon et al., 2016*), while the enhanced sensitivity of the Caatinga could indicate a potential relationship between vegetation cover and phenology with precipitation changes (*Barbosa, Huete & Baethgen, 2006*; *Seddon et al., 2016*).

Although most of the Brazilian fire-dependent and fire-sensitive biomes have high resilience to a gradual increase in climatic stress, and consequently have low vulnerability, Pampa, Atlantic Forest and the fire-independent biome are likely to have low resilience and be highly vulnerable (*Anjos & De Toledo, 2018*; *Pinho et al., 2020*). Consequently, we found that most fire-dependent biomes presented low climate risk, partially corroborating our risk hypothesis. However, the Cerrado biome presented a high risk, similar to that of the fire-independent biome (Caatinga). Finally, the fire-sensitive biomes presented opposite trends concerning risk: while the Amazon presented the lowest, the Atlantic Forest presented the highest risk among the Brazilian biomes.

We used relative differences between biomes and between fire-dependent, fire-independent, and fire-sensitive biomes to describe the overall patterns of climate risk, considering the standard deviation as a measurement of uncertainty. In this sense, our narrative is based on comparisons between biomes, which were ranked from lower to higher susceptibility considering the different metrics used (*e.g.*, hazard, resilience, vulnerability index). Although we acknowledge that such comparisons are mostly descriptive, the innovative combination of different metrics used in this study is a good starting point for future studies, which could delve into more robust analyzes focusing on each of the broader characteristics and overall patterns discussed here.

## Perspectives on Brazilian environmental policies

The development of fire and climate management programs to preserve the integrity of Brazilian biomes is paramount to reduce the most severe climate-related and fire-related impacts in these systems. Although the 2021 26th Conference of the Parties (COP26) of the United Nations Framework Convention on Climate Change (UNFCCC, a.k.a. "Climate Convention") made clear statements concerning our proximity to reaching dangerous tipping points for the climate system and the maintenance of the Amazon rainforest (*Walker, 2021*), Brazil's commitments to the convention goals were restricted to general propositions of banning illegal deforestation by 2030 and becoming carbon neutral by 2050, with no proposed action plans (*Fearnside, 2021*).

Since Brazil's public policies do not protect Brazilian biomes and the government fails to avoid deforestation, fires, and climate change, the country is constantly under economic pressure from international traders and consumers to prevent and decelerate environmental destruction (*Gibbs et al., 2015*). Such pressure occurs mainly in the form of conditions placed on imports of soy and beef, which are commodities related to the increase in deforestation rates (*Kehoe et al., 2019*; *Ferrante & Fearnside, 2021*). Protecting biodiversity should be a priority for Brazil for reasons that go beyond the country's interest in having access to international markets. Protected biodiverse landscapes create truly regenerative and sustainable systems, enhancing food production while preserving biodiversity (*Kremen, 2020*). The conservation of Brazilian biomes is of global importance

since the benefits from its ecosystem services are not only local. Thus, actions must be immediately strengthened to discourage and control intentional fires, stop deforestation, and fight climate change.

## CONCLUSIONS

The patterns of fire persistence and fire occurrence over time are related to human-induced fires and key drivers of burning are likely to be intensified across Brazilian biomes in the future, potentially increasing the magnitude of the fires and jeopardizing the integrity of the biomes. Although climate change is not necessarily the leading cause of the fires observed in fire-dependent, fire-independent, and fire-sensitive biomes, it is indeed likely to potentialize fire occurrence and spread by creating appropriate climate conditions in these biomes. Management actions should therefore prioritize programs to preserve the integrity of Brazilian biomes and reduce the most severe climate-related and fire-related impacts in these systems.

## ACKNOWLEDGEMENTS

We thank Alistair Seddon for providing the vegetation sensitivity data.

### Funding

This study was financed by the Coordenação de Aperfeiçoamento de Pessoal de Nível Superior - Brasil (CAPES) - Finance Code 001. Paulo Ferreira received financial support from Fundação para a Ciência e a Tecnologia (grants UIDB/05064/2020 and UIDB/04007/2020). Mariana M. Vale was funded by the Conselho Nacional do Desenvolvimento Científico e Tecnológico (CNPq) (Grant no. 304309/2018-4) and Fundação Carlos Chagas Filho de Amparo à Pesquisa do Estado do Rio de Janeiro (FAPERJ) (Grant no. E-26/202.647/2019) and had the support of the National Institute for Science and Technology in Ecology, Evolution and Biodiversity Conservation (CNPq Grant no. 465610/2014-5 and FAPEG Grant no. 201810267000023). Carlos Frederico Duarte Rocha was supported by FAPERJ through the program Cientistas do Nosso Estado (process E-26/202.803/2018) and from CNPq (Processes 302974/2015-6, 424473/2016 and 304375/2020-9), and by Programa Prociência from the Universidade do Estado do Rio de Janeiro (UERJ). Lilian Sales was supported by the Canadian Government through a Banting Postdoctoral Fellowship (FRN 174541). Paulo Ferreira was funded by the National Council for Research and Development (CNPq) (311103/2015-4), Foundation for the Support of Research of the State of Amazonas (FAPEAM) (01.02.016301.000289/2021-33), and National Institute for Research in Amazonia (INPA) (PRJ15.125). Mariana M. Vale and Paulo Ferreira developed this work in the context of the Brazilian Research Network on Climate Change (Rede Clima) (FINEP Grant no. 01.13.0353-00). The funders had no role in study design, data collection and analysis, decision to publish, or preparation of the manuscript.

## Grant Disclosures

The following grant information was disclosed by the authors:

Coordenação de Aperfeiçoamento de Pessoal de Nível Superior - Brasil (CAPES).

Fundação para a Ciência e a Tecnologia: UIDB/05064/2020, UIDB/04007/2020.

Conselho Nacional do Desenvolvimento Científico e Tecnológico (CNPq): 304309/2018-4.

Fundação Carlos Chagas Filho de Amparo à Pesquisa do Estado do Rio de Janeiro (FAPERJ): E-26/202.647/2019.

National Institute for Science and Technology in Ecology, Evolution and Biodiversity Conservation: 465610/2014-5.

FAPEG: 201810267000023.

FAPERJ through the program Cientistas do Nosso Estado: E-26/202.803/2018.

CNPq: 302974/2015-6, 424473/2016, 304375/2020-9.

Programa Prociência from the Universidade do Estado do Rio de Janeiro (UERJ).

National Council for Research and Development (CNPq): 311103/2015-4.

Foundation for the Support of Research of the State of Amazonas (FAPEAM): 01.02.016301.000289/2021-33.

National Institute for Research in Amazonia (INPA): PRJ15.125.

Brazilian Research Network on Climate Change (Rede Clima): 01.13.0353-00.

## Competing Interests

The authors declare there are no competing interests.

## Author Contributions

- Luisa Maria Diele Viegas conceived and designed the experiments, performed the experiments, analyzed the data, prepared figures and/or tables, authored or reviewed drafts of the article, and approved the final draft.
- Lilian Sales conceived and designed the experiments, performed the experiments, analyzed the data, prepared figures and/or tables, authored or reviewed drafts of the article, and approved the final draft.
- Juliana Hipólito performed the experiments, analyzed the data, prepared figures and/or tables, authored or reviewed drafts of the article, and approved the final draft.
- Claudjane Amorim analyzed the data, authored or reviewed drafts of the article, and approved the final draft.
- Eder Johnson de Pereira conceived and designed the experiments, performed the experiments, analyzed the data, authored or reviewed drafts of the article, and approved the final draft.
- Paulo Ferreira conceived and designed the experiments, performed the experiments, analyzed the data, authored or reviewed drafts of the article, and approved the final draft.
- Cody Folta performed the experiments, analyzed the data, prepared figures and/or tables, authored or reviewed drafts of the article, and approved the final draft.
- Lucas Ferrante analyzed the data, authored or reviewed drafts of the article, and approved the final draft.

- Philip Fearnside analyzed the data, authored or reviewed drafts of the article, and approved the final draft.
- Ana Claudia Mendes Malhado analyzed the data, authored or reviewed drafts of the article, and approved the final draft.
- Carlos Frederico Duarte Rocha analyzed the data, authored or reviewed drafts of the article, and approved the final draft.
- Mariana M. Vale conceived and designed the experiments, performed the experiments, analyzed the data, prepared figures and/or tables, authored or reviewed drafts of the article, and approved the final draft.

## Data Availability

The raw data is available in the Supplemental Files.

## Supplemental Information

Supplemental information for this article can be found online at http://dx.doi.org/10.7717/peerj.14276#supplemental-information.

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
