# Peer review of "We’re building it up to burn it down: fire occurrence and fire-related climatic patterns in Brazilian biomes"

_PeerJ, doi:10.7717/peerj.14276_

## Round 0.1 · original submission · Major Revisions

I have now received two independent reviews on your manuscript. Both reviewers were very positive about the content and goals of the study, while also providing suggestions on how to make the manuscript shorter and more straigtforward by re-oreganizing the order of some paragraphs in the mehods.
The introduction in indeed way too long and could be reduced. See R2 suggestions in that sense. It has 5 pages, try to keep it down to 3. For example, the explanation of resilience, vulnerability, and hazard can all be condensed to a single paragraph. The connection between paragraphs also has to be much improved. Many cited references are not in the ref list, double check that (e.g., Gonçalves-Souza et al. 2020, 2021). Discussion also needs to be very much reduced. There's no need to reiterate your hypotheses in the Discussion. I gave some guidance in the pdf attached.

While reporting ENMs, please follow these two protocols, while making the description as short and straightforward as possible:

https://onlinelibrary.wiley.com/doi/abs/10.1111/ecog.04960

https://www.nature.com/articles/s41559-019-0972-5

How are you able to make inferences about the difference in terms of resilience, risk etc among biomes without performing any explicit statistical analysis? All the data derived from the time series analysis in the Tables are descriptive and show as mean and SD. So how could you create a narrative, by for example ranking biomes in terms of vulnerability and resilience without addressing uncertainty?
You also mention the role of PAs in mediating biome vulnerability, but as far as I understood you haven't explicitely tested for it.

I have also made a few comments in the pdf attached. Please, reply to these comments as well in your rebuttal letter. Overall, I believe this is an important contribution to fire ecology in Brazil, and I'd like to see it published eventually, given authors are able to address reviewer's comments.

Reviewer 1 ·

Basic reporting

The manuscript is an excellent contribution to the knowledge of fire modeling and climatic changes. I think it can be published with some modifications.

Introduction: The introduction is too long I think this can be shortened. If two or three paragraphs can be taken out from the introduction, it can be more concise.

The Methods. – To my knowledge, the methods are correct and well applied. I think that the use of the formulas could be more explained. Please explain the meaning of all symbols used in your formulas.
Results: I think the captions of Figures 3 and 2 are inverted. Please correct. The caption of Figure 3 is poor and needs improvements. Please explain the meaning of the colors. In the methods, we can see how this was built, but I think is important to make it clear here that someone that is looking for a piece of quick information can understand the figure without reading the entire paper.

Discussion: is well written and I have no suggestions

Conclusion: Conclusion needs substantial improvement. From line 669 to line 681 many ideas are not conclusions of this study. Please take It out.

Experimental design

No comment

Validity of the findings

no comment

Reviewer 2 ·

Basic reporting

PeerJ

Title: We're building it up to burn it down: Fire occurrence and Fire-related climatic patterns in Brazilian biomes

General comments:

Intro: To my opinion the intro section is too long. Authors use too much space to explain: hazard, risk, vulnerability and resilience. This can surely be summarized into a single paragraph as it refers to well known approach of “risk assessment”. This can be explained in details in the methods section if you find necessary. I suggest the authors to reorganize the introduction into five-to six paragraphs as follows: 1) Risk assessments of natural hazards (e.g. fires) are both needed and provide crucial information on underlying causes; 2) Natural fires and climate and land use changes; 3) Different impact on ecosystems according to their evolutionary relationship with fires; 4) Explain how vulnerability, exposure, hazard and ability to cope with fires affect resilience; 5) present hypotheses (right them down, not just enumerate them); 6) potential applications and future directions.

Experimental design

Methods

I would prefer to start by describing Brazilian biomes, their characteristics and a brief historical data on fires. Then, what variables were used and why, including risk assessment. Then, how did you treat them, calculate indexes, etc… I’d prefer to have a specific topic for statistical analyses where you could explain how did you construct your models and tested them.

Validity of the findings

Results

The first paragraph pf the results (lines 421-423) are actually methods. Please remove from this section.

I it common the call figure in subtitles? Looks odd.

What do you mean by “occurrence rates”? Why not only “occurrence”? Is it so important to emphasize the rate? In the methods you do not use this word but to refer to sensitivity and specificity (line 333).

Line 431, replace narrow for small


Well, the results are hard to follow because they do not follow a sequence that the readers must be prepared to, before in the methods. I suggest the authors to use the same sequence of subtitles for the methods and results in order to keep a easy-to-follow flow of the results. Just to given and example, Figure 2 is called at the beginning and at the final paragraph. This figure is actually called everywhere in the results.

Discussion:

First phrase of discussion seems methods: “we evaluated…”. Please make a brief of your most important results.

Here it is important to use the same sequential logic that is also needed in the resutls. Theere is not a single word in the first paragraph of the discussion regarding risk, hazard, exposure, vulnerability and resiulience.

In the second paragraph the authors start by posing a hypothesis and explaining their expetations. I found this confusing because in the discussion we often use space to compare results with other finds and feedback theories.

Lines 526-530: This sentences must be used and highlighted in the first paragraph. This is your most important result and I sugget you to start the discussion with this.

The discussion on the potential effects of climate change on fires occurence among biomes is good but can be summarized in a little less words. I suggesto reducing it to a single paragraph.

Overall, discussion is too long and a bit hard to follow. Please consider reducing its lenght and adopt a logic sequence that follows the results presented in the results sections.

Conclusion

This can also be significantly reduced if you avoid to explain the work (at the very beggining) and avoid repeating results. This whole first paragraph can be deleted as the second one fits better as conclusions.

---

## Round 0.2 · Minor Revisions

Thank you for providing detailed responses to reviewer's and mine comments. The original reviewers didn't want to comment again on the text. The new version of the Introduction and Discussion are indeed much better. So, I'll make a few more suggestions:

1) The ODMAP protocol requres you to deposit a supplementary table with details of each modelling step. A template is available at http://www.ecography.org/appendix/ecog-04960

2) You need to separate the Suppl Tables into different files instead of having all as tabs in a single excel file

3) you need to include a couple of sentences in the Discussion about the shortcommings related to the lack of formal statistical analysis and elaborating a bit more from your response to my comment on the Rebuttal. Somehow you need to incorporate it so readers can understand your reasoning when reading the text.

---

## Round 0.3 · accepted · Accept

Thank you for making those final amendments to the manuscript. I'm happy to accept it as is and anticipate it'll be a great contribution to fire ecology in Brazil and hopefully inform efficient public policies.